# New Perspectives on the Effect of Dandelion, Its Food Products and Other Preparations on the Cardiovascular System and Its Diseases

**DOI:** 10.3390/nu14071350

**Published:** 2022-03-24

**Authors:** Beata Olas

**Affiliations:** Department of General Biochemistry, Faculty of Biology and Environmental Protection, University of Lodz, Pomorska 141/143, 90-236 Lodz, Poland; beata.olas@biol.uni.lodz.pl; Tel.: +48-42-6354485

**Keywords:** cardiovascular disease, dandelion, food product, safety

## Abstract

Cardiovascular diseases (CVDs) have been the leading cause of death for over 20 years. The main causative factors are believed to be high cholesterol, obesity, smoking, diabetes, and a lack of physical activity. One of the most commonly used treatments is a combination of anticoagulant and antithrombotic therapy; however, it often causes unwanted side effects. The European Society of Cardiology, therefore, recommends a prophylactic strategy, including a varied diet rich in fruits, vegetables, and medicinal plants; all of which are sources of natural compounds with antiplatelet, anticoagulant, or antioxidant activities, such as phenolic compounds. One such plant with multidirectional health-promoting effects and a rich source of secondary metabolites, including phenolic compounds, is dandelion (*Taraxacum officinale*). The present mini-review presents the current state of knowledge concerning the effects of dandelion consumption on the cardiovascular system and CVDs based on various in vitro and in vivo trials; it discusses the value of dandelion as a food product, as well as extracts and pure compounds, such as chicoric acid, which can be obtained from the various plant organs. The paper also sheds new light on the mechanisms involved in this activity and describes the cardioprotective potential of dandelion products and preparations.

## 1. Introduction

According to World Health Organization (WHO) reports, cardiovascular diseases (CVDs) have been the leading cause of death globally over the past 20 years. Despite the rapid development of innovative treatments and diagnostic techniques, it is estimated that around 20 million people worldwide are currently dying from CVDs [1]. The most common forms of CVD are atherosclerosis, arterial hypertension, myocardial infarction, stroke, and heart failure, with the main causative factors being high cholesterol, obesity, smoking, diabetes, and lack of physical activity. These symptoms are also typically accompanied by alterations in hemostasis.

Hemostasis is maintained primarily by a complex interaction between blood platelets, the endothelium, and various coagulation and fibrinolysis factors [2,3]. These processes play an important role by regulating the fluidity of circulating blood and by facilitating clot formation following damage to the endothelial wall [4]. However, hemostasis can be modulated by the effect of oxidative stress, resulting in the development of pathological processes that lead to the formation of atherosclerosis in acute coronary syndrome and cerebral ischemic events [5,6,7]. 

One of the most common treatment strategies employed for CVDs is based on a combination of anticoagulant and antithrombotic therapy, including such drugs as aspirin, clopidogrel (tienopyrin), rivaroxban warfarin, dabigatran, and apixaban. These drugs act through a range of mechanisms [8,9]; for example, aspirin prevents the production of thromboxane A_2_ (TXA_2_) by inhibiting the activity of cyclooxygenase in blood platelets, while clopidogrel blocks P2Y12 receptors on blood platelets. Unfortunately, the use of such medication often causes unwanted side effects in many patients, such as hemorrhagic and ischemic complications [8,10,11]. 

The European Society of Cardiology recommends the consumption of a varied diet rich in fruits, vegetables, or medicinal plants, which are sources of natural compounds with antiplatelet, anticoagulant, or antioxidant activity. Dandelion (*Taraxacum officinale*) is also a good source of secondary metabolites, such as phenolic compounds. The chemical content of dandelion has been detailed in a recent review, together with its varied antibacterial, anti-inflammatory, cytotoxic, and diuretic properties [12,13,14,15,16]. However, no paper has yet reviewed the prophylactic and therapeutic potential of dandelion and its bioactive components against CVDs, and their effects on the cardiovascular system in general. 

To address this gap, the present paper is intended as a mini-review of current literature examining the effects of dandelion on the cardiovascular system and its diseases in various in vitro and in vivo studies; it also describes the properties of its food products and preparations, as well as its extracts, complex fractions, and pure compounds, such as chicoric acid, obtained from the roots, leaves, fruits, and flowers. This paper also sheds new light on the mechanisms involved in their action and describes the cardioprotective potential (including antiplatelet and anticoagulant action) of these products and preparations from different plant organs. This review is based on papers identified in electronic databases: PubMed, Scopus, ScienceDirect, and Web of Knowledge. The last search was run on December 30 2021. The following terms were used: “dandelion;” “Taraxacum officinale;” “dandelion and cardiovascular disease;” “dandelion and hemostasis;” and “dandelion and oxidative stress.”

## 2. General Description of Dandelion and Its Products

The dandelion, Taraxacum officinale, also known as the common denichine or the drinifer, is a percentine of the Asteraceae family, a subfamily of the Cichorioideae. The Latin name of the plant derives from the Greek words taraxic, meaning ignition, and akeomia, i.e., treatment [17]. Although the dandelion is believed to originate in Europe, it is found throughout the entire northern hemisphere, including northern Europe, the temperate zone of North America, as well as Asia [15,17,18]. It can grow from sea level to alpine elevations and tolerates every soil type [12,13,14,15,16].

Dandelion is a rich source of phenolic acids (chicoric acid, chlorogenic acid), flavonoids (luteolin derivatives, quercetin), and terpenes (sesquiterpene lactones). It is also a strong source of vitamins (A, C, E, K, and B) and minerals (calcium, sodium, magnesium, iron, copper, silicon, zinc, manganese) [17,19]. More details about the chemical composition and nutritional value of dandelion plant organs are given by Grauso et al. [14], Lis and Olas [15], and Garcia-Oliveira et al. [16].

Although dandelion is mainly known for its medicinal properties, it has for many years been successfully used worldwide in the food industry as an entirely non-toxic and edible plant; indeed, the U.S. Food and Drug Administration has placed dandelion on the list of safe products for people with rare allergies [20,21]. However, the dose should not exceed 4 g or 12 g per day for the aerial parts of the plant or 1 g or 3 g per day for the root. The roots, leaves, and flowers may be eaten raw or cooked [14].

Due to their high nutrient content, dandelion leaves are often included as a salad ingredient, and the inulin-rich roots are used as substitutes for coffee or tea. In Asian countries, fried dandelion leaves are popular in combination with brown rice. Young leaves can also be prepared in the form of soup. In Turkey, fresh dandelion leaves are used as a spice and added to many dishes, and the ground dried leaves can also be used as a seasoning [22]. Dandelion leaves are also believed to have a positive effect on the cardiovascular system due to their high potassium content (397 mg potassium/100 g) [19,23]; indeed, increased potassium intake with food (about 3500 mg/day for an adult) has been found to lower blood pressure [23].

While both the flowers and roots can be used for winemaking, the two parts require different production processes [24]. In addition, the flowers, leaves, and roots can be consumed as herbal teas, while the flowers can also be made into syrup. In Canada and the UK, whole dandelions [25] are added to beer, while in Belgium, the flowers can be used as additives to a beer-based drink called saison, known for its strong fruity aftertaste. In addition, flower extracts can be used as flavor additives in a range of foods such as desserts, candies, baked cakes, jellies, and puddings [19]. The flower buds can be added to pancakes and omelets, and in some European countries, they are preserved in vinegar and served similar to capers [25]. Dandelion marmalades and liquors are commonplace in Italy [21]. In Great Britain, dandelion wine, made from petals and sugar and often lemon juice, is also popular as a carbonated drink.

Dandelion extracts can be purchased in capsule form as dietary supplements, especially in North America [15,26,27]. In addition, dandelion is a valuable species for supplying bees with nectar [18].

In addition to its flavor qualities, dandelion is traditionally used in infusions and decoctions as aperitif, tonic, and stimulant; however, studies suggest only a minor effect [17]. It has also been used for many centuries as a remedy for kidney, liver, and gallbladder disorders [27,28,29,30,31,32,33,34]. The beneficial effects of dandelion are dependent on the chemical compounds contained in the plant. These include sesquiterpene lactones, which have been found to have anti-inflammatory and antibacterial effects, as well as triterpenes or phytosterols, which possess anti-atherosclerotic properties. In addition, dandelions have high levels of phenolic compounds, including phenolic acids, with antioxidant properties and coumarins with anticancer, anti-inflammatory, antibacterial, and antithrombotic effects. The roots are also rich in inulin, which has a probiotic, hypoglycemic, and immune-boosting effect [14,15,17,19]. 

## 3. The Effect of Dandelion and Its Products on Cardiovascular System

### 3.1. In Vivo Studies

In vivo studies suggest that dandelion organs, especially the leaves and roots, may play an important role in the prophylaxis and treatment of CVDs [35,36,37,38,39,40,41,42,43,44]. 

#### 3.1.1. Antioxidant and Hypolipidemic Properties

Choi et al. [36] studied the effect of dandelion treatment on antioxidative enzyme and lipid profiles in rabbits consuming a high-cholesterol diet. Briefly, the rabbits were administered 1% dandelion leaf extract, containing 8% phenolic compounds, and 1% dandelion root extract, with 9% phenolic compounds; their cholesterol and total lipid levels were measured after administration. It was found that the dandelion treatment (250 g/day for one month) appeared to have antioxidant properties and resulted in improvements in aortal thickness. However, these results were not consistent with the beneficial effect on blood lipids in the context of a high-fat diet.

Another study by Majewski et al. [37] examined the effect of four-week supplementation with phenolic fractions from dandelion petals and leaves (enriched with L-chicoric acid) on Wistar rats. The rats were fed dandelion leaf or petal fractions daily, with 4.10 ± 0.05 and 1.41 ± 0.07 mg L-chicoric acid, respectively. Both dandelion fractions were found to have antioxidant properties, and the leaf fraction appeared to modify the lipid profile of the rats, resulting in decreased total cholesterol and triglyceride. The authors attribute this positive spectrum of activities to the chicoric acid present in the samples [37]. 

The oxidative stress plays a key role in the development of hypertension. Aremu et al. [38] observed that 21-day treatment with 500 mg/kg/day dandelion leaf extract (70% ethanol-water (*v*/*v*) extract, with 4 GA eq/mg extract total phenolic content) had antioxidant properties, which conferred significant antioxidant protection against hypertension in rats stimulated by free radical production: in this case, oxidative stress was induced by *N* ω-nitro- L-arginine methyl ester. The extract was found to elevate total antioxidant capacity and decrease lipid peroxidation, a marker of oxidative stress, in the heart, liver, kidney, and brain, among others. The authors also suggested that the phenolic compounds present in the tested extract regulate the level and activity of nitric oxide synthase (NOS) by interacting with kinase signaling pathways and the intracellular Ca^2+^ associated with NOS phosphorylation and NO production. In addition, these phenolic compounds might inhibit endothelin-1 (a vasoconstrictor) and endothelial NADPH oxidase. It was found that doses up to 1000 mg/kg BW did not cause mortality when administrated via the oral route, nor doses of 1600, 2900, and 5000 mg/kg BW. 

#### 3.1.2. Anti-Obesity Effect

Dandelion extract (150 and 300 mg/kg, 10 weeks) was found to have anti-obesity effects in mouse and rat models [39,40], manifested as a decrease in body weight among animals consuming a high-fat diet [40]; however, the chemical content of the extract and its total phenolic content were not specified in the study. Similarly, 60% ethanolic dandelion leaf extract, with a total phenolic content of about 123 mg gallic acid equivalent (GA) per g was also found to have anti-obesity properties [41]; treatment was found to improve the lipid profile and aspartate aminotransferase (AST) and alanine aminotransferase (ALT) concentrations in obese mice. The animals received this plant extract at 300 mg/kg body weight (BW)/day for eight weeks. In addition, docking analysis based on the computation of molecular-binding energies suggests that some of the phenolic compounds present in the extract, including kaempferol, luteolin, and myricetin, may inhibit pancreatic lipase activity.

Majewski et al. [42] indicated that consumption of dandelion flower syrup (27.82% for four weeks) did not appear to influence body weight when consumed with a normal-fat diet. However, the tested syrup was found to positively regulate antioxidant status and prostanoid content in an obese rat model. The primary phenolic components of this syrup were flavonoids (about 22 µg luteolin eq/mL) and hydroxycinnamic acids (about 468 µg L-chicoric acid eq/mL) [42]. 

It was also found that treatment with *T. coreanum* Nakai regulated lipid metabolism via the liver kinase B1–AMP-activated protein kinase signaling pathway, and promoted β-oxidation by a peroxisome proliferator-activated receptor [43]. As such, this plant may have a beneficial effect on obesity. In this experiment, male C57BL mice received 100 or 200 mg/kg of dandelion leaf extract or root extract orally for five weeks.

#### 3.1.3. Other Biological Activities

Modareri et al. [44] studied the effect of a 20-day course of hydro-alcoholic injections of dandelion extract (50–200 mg/kg body weight) on the levels of various blood cells, including blood platelets, in mice. However, no information was given about the plant organ used for the extract or its chemical content. 

Although dandelion preparations have been found to demonstrate multifunctional action on the cardiovascular system and CVDs (Table 1 and Figure 1), they have only been observed in animal models. There is a need for more observations on healthy people or people with various CVDs who have been supplemented by dandelion preparations.

### 3.2. In Vitro Study

Some papers indicate that preparations obtained from dandelion organs may have an effect on hemostasis, including blood platelet activation and coagulation processes in in vitro studies [45,46,47,48,49,50]. For example, an analysis of the effects on coagulation activity of human plasma found that 85% dandelion leaf fraction and 50% and 85% dandelion petal fractions significantly increased thrombin time in the range of 0.5–50 μg/mL. In addition, these fractions significantly increased plasma thrombin time when pre-incubated with thrombin at the highest concentration (50 μg/mL) [47,49]. In contrast, no anticoagulant effect was reported for dandelion fruit preparations incubated with human plasma in in vitro studies [45]. 

Lis et al. [45] studied the effects of three extracts from dandelion fruits on blood platelet activation; a methanolic extract of dandelion fruits (E1) and two extracts enriched with polyphenols (188 mg gallic acid equivalents/g); one with cinnamic acids (extract E2; 448 mg GA eq/g) and another with flavonoids (extract E3; 377 mg GA eq/g). The study also examined the activity of an additional four fractions isolated from extract E3: fraction A (luteolin fraction; 880 mg GA eq/g), fraction B (philonotisflavone fraction; 516 mg GA eq/g), fraction C (flavonolignan fraction; 384 mg GA eq/g), and fraction D (flavone aglycone fraction; 632 mg GA eq/g). The authors noted that extracts E2 and E3 and fractions A, B, and C significantly inhibited ADP-activated platelet adhesion to fibrinogen, while flavonoid fractions A to C inhibited thrombin-activated platelet adhesion at both tested doses (10 and 50 μg/mL). Other findings suggest that this inhibition may be related to low GPIIb/IIIa receptor exposition. In addition, extract E3 and the four fractions (A–D; 50 μg/mL) significantly inhibited the arachidonate metabolism in thrombin-activated platelets, with the luteolin-rich fraction A (10 and 50 μg/mL) inhibiting this process by approximately 60%. Similarly, the flavonoids present in dandelion fruits may also affect blood platelet activation by influencing the arachidonic acid metabolism [45]. Extracts E2, E3, and fraction A (50 μg/mL) were found to demonstrate anticoagulant potential against whole blood; however, this may be conditioned by the presence of caffeic acid derivatives (mainly chicoric acid), flavonoid derivatives, and luteolin. The effect of different dandelion preparations on blood clot formation was measured in a real-time hydrodynamic blood flow model using Total Thrombus-formation Analysis System (T-TAS); platelet thrombus formation was visualized on a collagen-coated chip.

Other studies have examined the effect of chicoric acid, the main component of the dandelion leaf and petal fractions, on blood platelet activation [48]. At two tested concentrations (10 and 50 μg/mL), chicoric acid inhibited thrombin-stimulated platelet adhesion to collagen by approximately 20%, and inhibited arachidonic acid metabolism. It was also found that the tested fractions from dandelion leaves and petals reduced the adhesion of resting platelets and thrombin-stimulated platelets to collagen. Moreover, chicoric acid (50 μg/mL) inhibited the adhesion of thrombin-activated platelets to fibrinogen by about 20%, and ADP-activated platelets by about 40%. 

In addition, the dandelion leaf and petal fractions also demonstrated anti-adhesive and anti-aggregatory effects [48] with the leaf fraction demonstrating stronger antiplatelet activity than chicoric acid alone, which may be due to the synergetic effect of the phenolic compounds within the fraction. In addition, the leaf fraction contains various phenolic acid and flavonoid derivatives, including luteolin, which may have a stronger effect than the single compound alone [48]. Elsewhere, treatment with dandelion leaf fraction was found to reduce P-selectin exposition and the presence of active GPIIb/IIIa on collagen-stimulated platelets and inhibit thrombus formation in human whole blood [50]. These parameters were measured by T-TAS and flow cytometry. 

Lis et al. [51] studied the effect of dandelion root extract and its various ingredients on the biological functions of human blood platelets. The experiment used five fractions (A–E) differing in chemical composition: fractions A and B contained 95% and 86% sesquiterpene lactones, fraction C contained mainly hydroxyphenylacetic acid derivatives (about 80% of all compounds), and the dominant constituents of fractions D and E were hydroxycinnamic acids. All tested dandelion root fractions (A–E) significantly inhibited the adhesion of resting blood platelets to collagen. Fraction C (50 μg/mL) and fractions D and E (10 and 50 μg/mL) demonstrated the strongest inhibition of thrombin-stimulated platelet adhesion to collagen, while fraction D (10 and 50 μg/mL) and fractions B, C, and E (50 μg/mL) significantly inhibited thrombin-induced platelet adhesion to fibrinogen. The greatest inhibitory effect on ADP-stimulated platelet adhesion to fibrinogen was observed for fraction E (10 and 50 μg/mL). 

Lis et al. [51] reported that fractions A, B, and C from dandelion roots (50 μg/mL) inhibited ADP-stimulated (10 μM) blood platelet aggregation, a risk factor for CVDs, by approximately 20% compared to controls. However, none of the tested root fractions significantly influenced platelet aggregation stimulated by 2 μg/mL collagen [51], nor 1 U/mL thrombin [49]. 

These results may suggest that the tested dandelion root preparations may exert their anti-aggregation activity by interacting with ADP receptors on the blood platelet membrane [51]. However, the fractions did not influence the exposition of active GPIIb/IIIa on ADP-activated (10 and 20 µM) blood platelets in whole blood [50]. Furthermore, none of these fractions appeared to affect the Platelet Reactivity Index (PRI) [50], a parameter known to correspond with vasodilator-stimulated phosphoprotein (VASP) phosphorylation level. 

Lis et al. [51] also found that dandelion root fractions A–E (10 and 50 μg/mL) did not appear to have any significant effect on arachidonic acid metabolism, resulting in the formation of thromboxane A_2_. However, the authors noted that the best antiplatelet activity was demonstrated by fraction C, characterized by high levels of hydroxyphenylacetic derivatives of inositol and chlorogenic acids. In addition, as the other phenolic components, including the hydroxycinnamic acid derivatives in fractions D and E, were not found to display such high activity, the authors suggested that sugar alcohol (inositol) may be responsible for the observed antiplatelet properties [51]. Moreover, fractions D and E demonstrated a stronger anticoagulant effect than other fractions (A, B, and C) [49]; it was hypothesized that the anticoagulant properties of these fractions, as well as the underlying mechanism, were associated with the modulation of thrombin activity.

It is important to note that none of the tested dandelion extracts or fractions, nor the chicoric acid isolated from various dandelion organs (0.5 to 50 μg/mL) caused blood platelet lysis, i.e., lactate dehydrogenase leakage, into the extracellular environment [45,47,48,49,51]. 

The antioxidant performance of various dandelion preparations was assayed based on lipid peroxidation level, defined as thiobarbituric acid reacting substances (TBARS), and protein oxidation level, based on thiol and carbonyl group level, using plasma and blood platelets: two important elements of hemostasis. Some results have shown that dandelion leaf, petal, and fruit preparations have antioxidative potential [45,47]. For example, lipid peroxidation in human blood platelets treated with H_2_O_2_/Fe^2+^ was inhibited by exposure to a 50% dandelion leaf fraction and a 50% dandelion petal fraction (both 50 μg/mL) [47]; in addition, after 30 min of incubation with a 50% leaf fraction (1 and 50 μg/mL), the platelet proteins demonstrated elevated thiol group numbers compared to controls. However, the tested fractions did not demonstrate significant changes in protein carbonylation in blood platelets [47]. Jedrejek et al. [46] indicated that phenolic fractions from petals have better antioxidant activity than those from leaves in human plasma; they propose that this may be due to the fact that the petals are better sources of flavonoids than the leaves. 

Lis et al. [45] reported that three of the tested extracts (E1–E3) and two of the luteolin-rich fractions of dandelion fruits (fraction A) and flavone aglycones (fraction D) inhibited H_2_O_2_/Fe^2+^-stimulated lipid peroxidation in human plasma when administered at 50 μg/mL; however, no such antioxidant effects were observed for dandelion fruit preparations containing philonotisflavones (fraction B) or flavonolignans (fraction C). In addition, extracts E2 and E3 and fractions B, C, and D inhibited the oxidation of plasma thiol groups following H_2_O_2_/Fe^2+^ treatment. All the tested dandelion fruit preparations (50 µg/mL) inhibited the carbonylation of proteins in human plasma treated with H_2_O_2_/Fe^2+^ [45].

Chicoric acid, the main component of dandelion fruit extracts and dandelion leaf and petal fractions, was also found to demonstrate antioxidant activity in human plasma and human blood platelets [48]. 

In addition, dandelion root fractions A–E were found to reduce the level of human plasma lipid peroxidation caused by H_2_O_2_/Fe^2+^, which generated hydroxyl radicals. This molecule has an unpaired electron in its outer orbit, and hence is a highly-reactive particle that can irreversibly damage biomolecules by forming chemical bonds with them. While all fractions (A–E) were found to demonstrate antioxidant effects at 5 μg/mL, both the C fraction (rich in hydroxyphenylacetic derivatives of inositol and chlorogenic acids) and the E fractions (rich in hydroxycinnamic tartaric acid derivatives) were also active at lower concentrations (1 μg/mL). In contrast, while the strongest effect of all tested fractions was reported at a concentration of 50 μg/mL, the best results were obtained for fractions C and for fractions A and E, these being a reduction of more than 30% compared to the positive control plasma treated with H_2_O_2_/Fe^2+^ alone [49]. 

Jedrejek et al. [49] also noted that all test fractions from dandelion roots (A–E) had a protective effect on thiol groups in human plasma proteins under oxidative stress; however, the best activity at concentrations of 0.5–50 μg/mL was demonstrated by fraction A, which was rich in amino acid derivatives of sesquiterpene lactones. In contrast, fraction B (containing sesquiterpene lactones and hydroxyphenylacetic inositol derivatives) displayed a protective effect at concentrations of 10 and 50 μg/mL, and fraction D (containing chlorogenic acid) at doses of 0.5, 1, 10, and 50 μg/mL. While all tested dandelion root fractions inhibited H_2_O_2_/Fe^2+^-induced plasma protein carbonylation, this change was not always statistically significant [49]. 

Dandelion root fractions A–E also significantly inhibited H_2_O_2_/Fe^2+^-induced lipid peroxidation in human blood platelets. More precisely, 70% inhibition was observed for all five fractions (A–E) at a concentration of 50 μg/mL; however, only A and E demonstrated a protective effect on protein thiol groups at the lowest test concentration (10 μg/mL). Fraction C inhibited protein carbonylation at two tested concentrations (10 and 50 μg/mL), while fraction A was only active at the highest dose (50 μg/mL) and fraction D at the lower concentration (10 μg/mL) [51]. In addition, only fraction A (10 μg/mL) had a significant effect on reducing the level of superoxide anion (O_2_^•−^) in resting blood platelets [51].

Kontgiorgis et al. [52] studied the optimum preparation conditions of *T. officinale* beverage with respect to its antioxidant properties. They used dried, commercial dandelion and prepared beverages boiling for 1, 3, and 5 min. Samples prepared under 3 min boiling had the best antioxidant activity, for example, they inhibited lipid peroxidation of linoleic acid. 

## 4. Conclusions

Conventional antiplatelet or anticoagulant drug therapy is often associated with side effects, and in such cases, the use of natural compounds may be a safer alternative. Recent studies have evaluated the potential of various plants to modulate hemostasis with the aim of using them as potential candidates for the prophylaxis and treatment of CVDs. The present study gathers a number of in vitro and in vivo studies that examine the effect of dandelion and its products on various elements of hemostasis, including blood platelets. Studies have found that the active components of dandelions are able to modulate hemostatic processes, including the signal pathways associated with blood platelets, in varied and sometimes opposing ways. Some of the hemostatic mechanisms exerted by dandelion and its products, and their effects on CVDs, are given in Figure 1 and Figure 2. Three key routes by which dandelions and their products may act are: (1) the inhibition of reactive oxygen species (ROS) production; (2) inhibition of the arachidonic acid metabolism; (3) reduction in the exposition of receptors on the platelet surface. 

Although the effects of dandelion and its products on hemostasis have been evaluated in different in vitro and in vivo studies (Table 1), the evidence base is insufficient to unequivocally confirm whether dandelion and its products have beneficial effects on hemostasis and CVDs, especially in humans. In addition, their effects on human hemostasis have only been evaluated in vitro, and the precise prophylactic and treatment doses are currently unknown. Both the immediate ingested context and the broader dietary context affect the potential of constituents of a food, beverage, capsule, or tablet to be digested, absorbed, and metabolized; the effect of the whole may differ substantially from the sum of the effects of the individual constituents of dandelion and its products. Therefore, more randomized clinical trials with larger groups are needed, especially including both healthy people and those with risk factors for CVD, including obesity, high cholesterol, smoking, diabetes, and a sedentary lifestyle.

## Figures and Tables

**Figure 1 nutrients-14-01350-f001:**
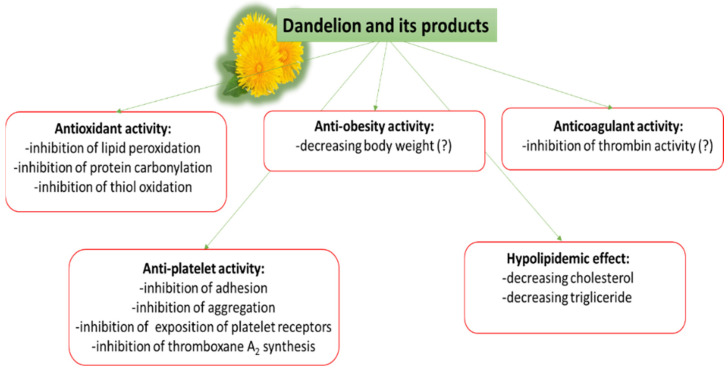
Multifunctional action of dandelion and its products on CVDs.

**Figure 2 nutrients-14-01350-f002:**
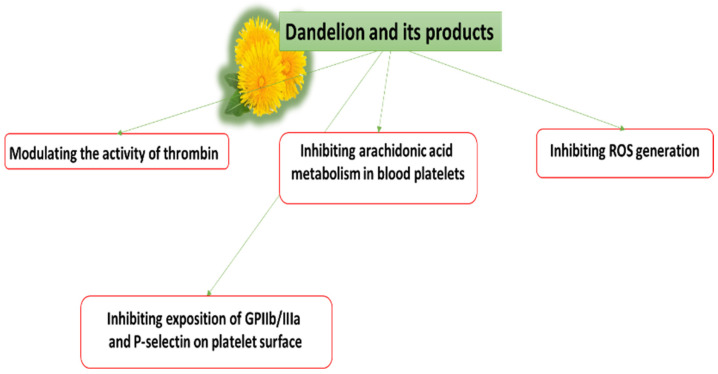
Proposed mechanisms of action of dandelion and its products on hemostasis.

**Table 1 nutrients-14-01350-t001:** Effect of dandelion preparations on the cardiovascular system and its diseases (in vivo and in vitro studies).

Dandelion Preparation	Dose	Subject	Effect	Reference
In vivo studies				
Flowers				
Water syrup	27.82% for 4 weeks	Obese rats	Antioxidant effect	[42]
Leaves				
1% extract	250 g/day for 1 month	Rabbits	Antioxidant effect and hypolipidemic properties	[36]
95% ethanol extract	400 mg/kg	Mice	Anti-obesity effect	[39]
Ethanolic extract	150 and 300 mg/kg for 10 weeks	Rats	Anti-obesity effect	[40]
60% ethanolic extract	300 mg/kg body weight (BW)/day for 8 weeks	Obese mouse	Anti-obesity effect	[41]
Phenolic fraction	694 mg/kg of diet for 4 weeks	Wistar rats	Antioxidant effect	[37]
70% ethanol-water (*v*/*v*) extract	500 mg/kg/day for 21 days	Hypertensive rats	Antioxidant effect	[38]
Petals				
Phenolic fraction	694 mg/kg of diet for 4 weeks	Wistar rats	Antioxidant effect	[37]
Roots				
1% extract	250 g/day for 1 month	Rabbits	Antioxidant effect and hypolipidemic properties	[36]
In vitro studies				
Fruits				
Flavonoid preparations: extracts and fractions	10 and 50 µg/mL	Human plasma and blood platelets	Antioxidant and antiplatelet effects	[45]
Leaves				
Phenolic fraction	1–50 µg/mL	Human plasma and blood platelets	Antioxidant and anticoagulant effects	[46,47]
Petals				
Phenolic fraction	1–50 µg/mL	Human plasma and blood platelets	Antioxidant and anticoagulant effects	[46,47,48]
Roots				
Preparations	0.5–50 µg/mL	Human plasma and blood platelets	Antioxidant, antiplatelet, and anticoagulant effects	[47,48,49]

## Data Availability

Not applicable.

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
