# Peer review of "New Perspectives on the Effect of Dandelion, Its Food Products and Other Preparations on the Cardiovascular System and Its Diseases"

_nutrients, 2022, doi:10.3390/nu14071350_

Round 1
Reviewer 1 Report
The paper entitled: “ New perspectives on the effect of dandelion, its food products and other preparations on the cardiovascular system and its diseases“ by Beata Olas* et al presents a really interesting overview of dandelion effect on cardiovascular diseases.
Major Comments
Paragraph “3 The effect of dandelion and its products on cardiovascular system”, needs to be further subcategorized according to the biological activity in each one of the provided two categories
Most of the paragraphs present mixed experiments and it is difficult for the reader to follow a specific series of results from similar experiments by different research groups
“Methodology” section, is missing, where the author could give us more information about the period of his literature search and the inclusion and exclusion criteria for the selected manuscripts.
The authors have not done extensively research on the literature and some studies are missing:” Antioxidant Profile of Home Prepared Taraxacum Officinale Weber Ex Wigg Beverage, DOI:10.2174/2665978601666200212110603” “H. J. Jeon et al., “Anti-inflammatory activity of Taraxacum officinale,” J. Ethnopharmacol., 2008.”, “E. Yarnell and K. Abascal, “Dandelion (Taraxacum officinale and T mongolicum).,” Integr. Med. A Clin. J., 2009.” and others
At the Conclusions, the authors are referring to clinical trials, although the tested extracts of dandelion are totally different and there are no information for a titrated extract used in the studies.
Minor Comments
Line 92, space need to be eliminated
Line 101, space need to be eliminated
Line 135, needs to be rewritten as follows: “In vivo studies”
Line 206, needs to be rewritten as follows: “In vitro studies”
Line 172, needs to be rewritten “It has known that oxidative stress plays a key role in the development of hypertension.”
Author Response
The paper entitled: “ New perspectives on the effect of dandelion, its food products and other preparations on the cardiovascular system and its diseases“ by Beata Olas* et al presents a really interesting overview of dandelion effect on cardiovascular diseases.
I would like to thank the Reviewer for providing helpful comments.
Major Comments
Paragraph “3 The effect of dandelion and its products on cardiovascular system”, needs to be further subcategorized according to the biological activity in each one of the provided two categories. Most of the paragraphs present mixed experiments and it is difficult for the reader to follow a specific series of results from similar experiments by different research groups
Response: I have added subcategorized: Antioxidant and hypolipidemic properties; Anti-obesity effect; and Other biological activities.
“Methodology” section, is missing, where the author could give us more information about the period of his literature search and the inclusion and exclusion criteria for the selected manuscripts.
Response: I have added this information: “This review is based on papers identified in electronic databases: PubMed, Scopus, ScienceDirect, and Web of Knowledge. The last search was run on December 30 2021. The following terms were used: “dandelion”, “Taraxacum officinale”, “dandelion and cardiovascular disease”, “dandelion and hemostasis”, and “dandelion and oxidative stress”.”
The authors have not done extensively research on the literature and some studies are missing:” Antioxidant Profile of Home Prepared Taraxacum Officinale Weber Ex Wigg Beverage, DOI:10.2174/2665978601666200212110603” “H. J. Jeon et al., “Anti-inflammatory activity of Taraxacum officinale,” J. Ethnopharmacol., 2008.”, “E. Yarnell and K. Abascal, “Dandelion (Taraxacum officinale and T mongolicum).,” Integr. Med. A Clin. J., 2009.” and others
Response: I have added more information from these papers:
Jeon, H.J., Kang, H.J., Jung, H.J., Kang, Y.S., Lim, C.J., Kim, J.M., Park, E.H., 2008. Anti-inflammatory activity of Taraxacum officinale” J. Ethnopharm. 11, 82-88.
Yarnell, E., Abascal, K., 2009. Dandelion (Taraxacum officinale and T mongolicum). Integr Med 8, 35-38.
Kontogiorgis, Ch., Deligiannidou, G.E., Karamani, V., Hodjipavlou-Litina, D., Lazari, D., Papadopoulos, A., 2020. Antioxidant profile of home prepared Taraxacum officinale Weber Ex Wigg beverge. Curr. Nutraceut. 1, 64-72.
Garcia-Oliveira, P., Barral, M., Carpena, M., Gullon, P., Fraga-Corrol, M., Otero, P., Prieto, M.A., Simal-Gandara, J. 2021. Traditional plants from Asteraceae family as potential condidates for functional food industry. Food Funct. 12, 2850-2873.
At the Conclusions, the authors are referring to clinical trials, although the tested extracts of dandelion are totally different and there are no information for a titrated extract used in the studies.
Response: I have added more information about it: “In addition, their effects on human hemostasis have only been evaluated in vitro, and the precise prophylactic and treatment doses are currently unknown. Both the immediate ingested context and the broader dietary context affect the potential of constituents of a food, beverage, capsule or tablet to be digested, absorbed, and metabolized; and the effect of the whole may differ substantially from the sum of the effects of the individual constituents of dandelion and its products. Therefore, more randomized clinical trials with larger groups are needed, especially including both healthy people and those with a risk factors for CVD, including obesity, high cholesterol, smoking, diabetes, and a sedentary lifestyle.”
Minor Comments
Line 92, space need to be eliminated
Response: I have corrected.
Line 101, space need to be eliminated
Response: I have corrected.
Line 135, needs to be rewritten as follows: “In vivo studies”
Response: I have corrected.
Line 206, needs to be rewritten as follows: “In vitro studies”
Response: I have corrected.
Line 172, needs to be rewritten “It has known that oxidative stress plays a key role in the development of hypertension.”
Response: I have corrected. Now, it is: “The oxidative stress plays a key role in the development of hypertension.”
Reviewer 2 Report
I’ve read with attention the review by Beata Olas, that is interesting, well-organized, overall well-written and update. I've only some minor comments:
- The narrative nature of the review should be stressed all over the text (starting from the title)
- The introduction of the abstract is too much long and not focused on the main focus of the paper
- The text should be revised by a native english speaker. In particular, there are some words that does not match with the sense the author tries to give. For instance, "Dandelion and its products" is not correct. Presumably the author mean "Dandelion and its bioactive components"
- The references are not strictly reported in the style of the journal
Author Response
I’ve read with attention the review by Beata Olas, that is interesting, well-organized, overall well-written and update. I've only some minor comments:
I would like to thank the Reviewer for providing helpful comments.
- The narrative nature of the review should be stressed all over the text (starting from the title)
Response: I have corrected.
The introduction of the abstract is too much long and not focused on the main focus of the paper
Response: I have corrected.
- The text should be revised by a native english speaker. In particular, there are some words that does not match with the sense the author tries to give. For instance, "Dandelion and its products" is not correct. Presumably the author mean "Dandelion and its bioactive components"
This paper has been proofread by a native speaker of English (certificate as attachment)
- The references are not strictly reported in the style of the journal
Response: I have corrected.